# Fast Treatment Personalization with Latent Bandits in Fixed-Confidence Pure Exploration

**Newton Mwai**                                                                    *mwai@chalmers.se*
*Department of Computer Science and Engineering*
*Chalmers University of Technology*

**Emil Carlsson**                                                                  *caremil@chalmers.se*
*Department of Computer Science and Engineering*
*Chalmers University of Technology*

**Fredrik D. Johansson**                                                  *fredrik.johansson@chalmers.se*
*Department of Computer Science and Engineering*
*Chalmers University of Technology*

**Reviewed on OpenReview:** *https://openreview.net/forum?id=NNRIGE8bvF*

## Abstract

Personalizing treatments for patients often involves a period of trial-and-error search until an optimal choice is found. To minimize suffering and other costs, it is critical to make this process as short as possible. When treatments have primarily short-term effects, search can be performed with multi-armed bandits (MAB), but these typically require long exploration periods to guarantee optimality. In this work, we design MAB algorithms which provably identify optimal treatments quickly by leveraging prior knowledge of the types of decision processes (patients) we can encounter, in the form of a latent variable model. We present two algorithms, the Latent LP-based Track and Stop (LLPT) explorer and the Divergence Explorer for this setting: fixed-confidence pure-exploration latent bandits. We give a lower bound on the stopping time of any algorithm which is correct at a given certainty level, and prove that the expected stopping time of the LLPT Explorer matches the lower bound in the high-certainty limit. Finally, we present results from an experimental study based on realistic simulation data for Alzheimer's disease, demonstrating that our formulation and algorithms lead to a significantly reduced stopping time.

## 1 Introduction

There is growing interest in using machine learning for personalizing medical treatments to account for heterogeneity in patients' responses. Finding a suitable choice for an individual often involves a phase of trial and error before settling on a therapy that works for them, especially in the treatment of chronic diseases (Fraenkel et al., 2021; Stern, 2009). In rheumatoid arthritis, for example, when first and second-line treatment fails, there is large variability in the choice of next therapy, and several drugs may be considered equally good choices a priori (Zink et al., 2001). Further, switching therapies has associated costs: every time a therapy is changed, the patient has to get used to the new therapy and its potential side effects. It is therefore desirable to minimize such switches, even if changes are to other equally good treatments after a treatment has been identified in the search phase. Learning algorithms could improve the efficiency of this search, reducing the number of avoidable trials (Chakraborty and Moodie, 2013).

A classical framework for exploring alternative treatments is Multi-armed Bandits (MAB) (Gittens and Dempster, 1979; Lai and Robbins, 1985), originally motivated by reducing suffering in drug testing (Thompson, 1933). However, MABs tend to be sample-hungry to the point of being unsuitable for finding personalized treatments in real-world clinical settings. Because a long search phase can prolong unnecessary suffering, it

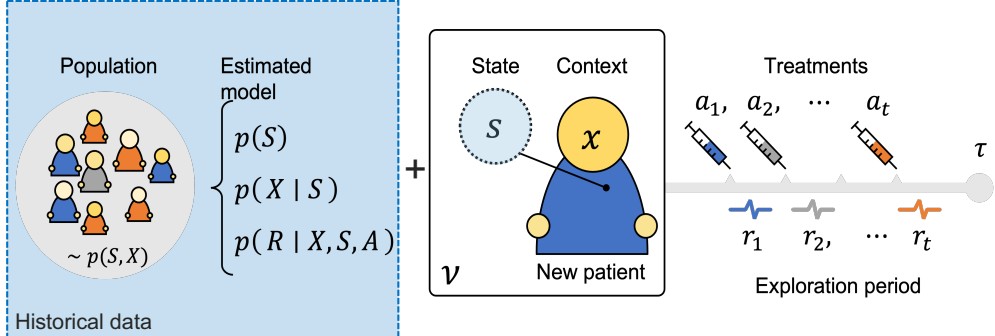

Figure 1: Illustration of the pure-exploration latent bandit problem and the example of treatment personalization. A population of patients have been observed in historical data to learn the distribution of latent states $P(S)$, $P(X|S)$ and the conditional reward the distribution $P(R|X, S, A)$. A new patient, represented by the instance $\nu = (x, s)$ is treated with actions $a_t$, observing rewards $r_t$ until the stopping time $\tau$.

must be avoided and minimized whenever possible. Existing methods for the fixed-confidence pure exploration setting in MABs, which aim to minimize the time it takes to find an optimal treatment at a given certainty level (Even-Dar et al., 2006; Garivier and Kaufmann, 2016; Russo, 2016; Shang et al., 2020) also yield long exploration phases.

One reason for the long exploration of bandit algorithms is that each instance—each patient, in our example—is treated as independent, learning parameters from scratch each time. This allows for complete personalization, often incorporating contextual or side information (Li et al., 2010; Chu et al., 2011), but disregards any similarities between instances. For many conditions, differences in responses (rewards) to treatment between patients are believed to be explained by a small number of disease subtypes (Devi and Scheltens, 2018; Borish and Culp, 2008). Thus, for a patient with a known subtype, an optimal treatment could be identified from the treatment responses of previous patients with the same subtype.

The subtype of a patient may be viewed as a latent state, as it is unobserved at the start of treatment, but manifests in a patient's responses to different therapies. With access to data on the treatment of previous patients, it is possible to fit a model of the distribution of latent states and their association with actions and rewards, for instance with variational inference methods (Kingma and Welling, 2013; Jang et al., 2016). Given such a model, for a new patient (bandit instance), our task becomes to identify which latent state they belong to, see Figure 1. Latent Bandits and recent iterations formalize this idea but are limited to regret minimization, aiming to minimize the regret compared to optimal actions over a possibly infinite period (Maillard and Mannor, 2014; Zhou and Brunskill, 2016; Hong et al., 2020a;b; Kwon et al., 2021). This differs from our goal of finding the optimal treatments within a desirably short exploration period, while also ensuring that the algorithm commits to a good treatment after exploration, without treatment switches.

In this work, we derive fixed-confidence pure-exploration bandit algorithms which aim to minimize the number of trials required to find an individual-optimal treatment by incorporating existing knowledge of latent structure.

**Main contributions. 1)** We propose a formulation of the personalized treatment search problem with known latent structure in the fixed-confidence pure-exploration setting (Section 2). **2)** We prove a lower bound for the search time of any algorithm in our latent bandit setting and prove a matching upper bound for the Latent LP-based Track and Stop (LLPT) Explorer (Section 3, 5). **3)** We present two algorithms, the LLPT Explorer and the Divergence Explorer (Section 4). **4)** We perform an extensive empirical evaluation on a simulator of Alzheimer's disease and illustrate that our formulation and algorithms lead to a significantly reduced stopping time compared to classical pure-exploration algorithms in the MAB framework (Section 6).

## 2 Problem formulation

We think of a treatment personalization strategy as an agent which interacts with a patient over $t \in \mathbb{N}$ rounds, aiming to try as few treatments as possible before the best possible treatment has been identified with a confidence level of at least $1 - \delta$, for a pre-specified $\delta > 0$. At the start of the sequence, the agent observes a patient's context covariates (e.g., lab measurements) as a draw of a random variable $X^1 \in \mathbb{R}^d$. Then, at each step $t = 1, 2, ...$, the agent takes an action $A_t \in \mathcal{A} = \{1, ..., K\}$ (trial treatment) and gets a reward $R_t \in \mathbb{R}$ (treatment outcomes). When an optimal treatment has been found, exploration stops and the agent recommends this treatment. The setting is illustrated in Figure 1. In the multi-armed bandit literature this setting is called fixed-confidence pure exploration (Garivier and Kaufmann, 2016; Shang et al., 2020).

A fixed-confidence pure-exploration strategy $\phi$ comprises a sampling rule for exploring actions $A_t$ at each step $t$, a stopping rule to decide the time $\tau$ at which the exploration is over, and a recommendation rule which returns the best action $\hat{a}_\tau$ at the stopping time $\tau$. Our goal is to design a strategy $\phi$ to minimize the expected stopping time $\mathbb{E}[\tau]$. In our healthcare example, this serves to minimize the search for optimal treatments, and thus minimize patient suffering in the treatment search phase while also ensuring that the algorithm commits to a good treatment after exploration, without treatment switches.

Even for state-of-the-art pure exploration algorithms, the necessary exploration tends to be long in realistic settings (see Figure 2). To overcome this, we will make structural assumptions about contexts, actions and rewards regarding patient similarity. In our healthcare example, it is plausible that a new patient (bandit problem instance) shares significant similarity with historical patients (logged bandit data), and that the optimal treatment for them is the same as for similar patients. However, in many domains, the context $X$ is not sufficient to identify optimal treatment since it does not account for all individual variation (Håkansson et al., 2020). To account for remaining individual variation between patients with the same $X$, we will assume that there is a finite number of latent states $S \in \mathcal{S} = \{1, ..., M\}$, e.g., patient types, which cannot be directly observed. Thus, the optimal treatment is determined by the context $X$ and the latent state $S$: two instances (e.g., two patients) are similar if they have the same context and latent state (e.g., disease subtype).

Identifying the true latent state $S$ is sufficient but not strictly necessary to solve our problem. For successful treatment, we are only interested to identify the optimal treatment at exploration stop, $\hat{a}_\tau$. Therefore, it is not necessary to estimate the correct latent state, but the set of latent states that have the same optimal arm. Having context $X$ is desirable as it helps reduce the number of trials if it is informative of the underlying latent state $S$, with unexplained variation further discoverable by trying different treatments.

A latent variable model (LVM) of the distribution of latent states $S$, contexts $X$, actions $A$ and rewards $R$ can be estimated from historical data and used to speed up exploration for a new subject. Maillard and Mannor (2014) and Hong et al. (2020a) made use of LVMs for "Latent Bandits" in the related setting of regret minimization. As these algorithms do not come with stopping and/or recommendation rules, they are not applicable to the fixed-confidence setting where the goal is to terminate search as quickly as possible.

In the MAB formalism, our problem can be defined as fixed-confidence pure-exploration latent bandits with a single initial context. In doing so, we assume that the latent subtype and the distributions of rewards is unaffected by time and previous actions. This is plausible for conditions treated with symptomatic therapies, such as for chronic degenerative disease like AD or Rheumatoid Arthritis (RA), where treatments typically target the symptoms and not the underlying disease pathology (Fish et al., 2019). Under these assumptions, the optimal choice of treatment remains fixed through exploration.

### 2.1 Fixed-confidence pure-exploration latent bandits

Given a state $s$, a context $x$, and an action $a$, let

$$\mu_{a,x,s} := \mathbb{E}[R \mid A = a, X = x, S = s]$$

---

[1] By convention, we use capital letters for random variables and lowercase for observed variables

denote the expected reward for that action, and let

$$\mu_{x,s}^* = \max_a \mu_{a,x,s} \quad \text{and} \quad a_{x,s}^* = \arg\max_a \mu_{a,x,s}$$

denote, respectively, the optimal expected reward and arm in latent state $s$ and observed context $x$. We assume that the maximizer $a_{x,s}^*$ is a single action for each state-context pair $(x, s)$, but our arguments can be generalized to the case with multiple optimal actions. Further, let $H_t = (X, A_1, R_1, ..., A_t, R_t)$ denote the history of context, actions and rewards, up to time $t$, letting $H_0 = (X)$. The utility of the context $X$ is in computation of the likelihood $P(s|H_t)$ and this is agnostic of either finite or infinite context assuming that a good model of the likelihood is known.

Our goal is to design a search strategy $\phi$ *to minimize the expected number of trials $\tau$ required to identify an optimal action, with confidence at least $1 - \delta$, for new subjects with context $X$ and unknown latent state $S$.*

$$\begin{aligned}
&\underset{\phi}{\text{minimize}} & & \mathbb{E}_{\phi,S,H_\tau}[\tau] & & (1) \\
&\text{subject to} & & P(\mu_{\hat{a}_\tau,x,s} < \mu_{x,s}^* \mid X = x, S = s) \leq \delta, \ \ \forall x, s
\end{aligned}$$

We say that a search strategy $\phi$ is $\delta$-PAC if the error probability is bounded by $\delta$. Here, this is captured by our constraint, $\forall x, s : P(\mu_{\hat{a}_\tau,x,s} < \mu_{x,s}^* \mid X = x, S = s) \leq \delta$, as long as the probability model is correct.

In equation 1, we minimize the expected stopping time (e.g., over a population of patients) while satisfying instance-dependent constraints (per patient). We justify this formalization by noting that, in our running example, any single patient will have a single random stopping time, which we can estimate and analyze only in expectation. However, it is desirable and possible to guarantee, per patient, that our confidence exceeds $1 - \delta$ whenever we stop.

We assume that a model $\mathcal{M}_\theta = \{p_\theta(S), p_\theta(X \mid S), p_\theta(R \mid A, X, S)\}$ of the marginal state probability $p(S)$ and the likelihood of observed variables under $S$, including the set of reward means $\mu_{a,x,s}$, is *available when search begins*, akin to Hong et al. (2020a). This means that once $s$ is known, so is the optimal arm in $s$, and no further exploration is necessary. Such a model can be learned from logged bandit instances, for example, using a variational autoencoder (Kingma and Welling, 2013), but this is outside the scope of this work.

For simplicity, we will assume that all reward distributions are stationary in time and Gaussian with equal variance $\sigma^2$, that is, given $A_t = a, X = x, S = s$, for all $t$

$$R_t \sim \mathcal{N}(\mu_{a,x,s}, \sigma^2) \ .$$

The algorithms presented in Section 4 are applicable in the non-Gaussian case as well, assuming that the reward distribution is known through $\mathcal{M}_\theta$, but our analysis in Section 5 is limited to Gaussian rewards for now. Our analysis makes heavy use of the Kullback-Leibler (KL) divergence, and we will adopt the notation $\text{KL}(\mu_{a,x,s} \| \mu_{a,x,s'}) = \text{KL}(p(R \mid a, x, s) \| p(R \mid a, x, s'))$ for the KL-divergence between the two Gaussian rewards for arm $a$ under states $s, s'$ with equal variance $\sigma^2$ and means as indicated.

## 3   Lower bound on stopping time

To serve as benchmark for our algorithms, we derive a lower bound on the worst-case solution to objective equation 1 for any algorithm which satisfies its constraints.

The seminal work of Kaufmann et al. (2016) presented a general inequality from which one can derive lower bounds for $\delta$-PAC algorithms in the best-arm identification framework. In lemma 1, we present a variant of their key result, adapted to our latent bandit setting. For brevity, we let

$$\rho(x; s, s') = \log[p(x \mid s)/p(x \mid s')]$$

denote the log-likelihood ratio of the observed context $x$ under latent states $s$ and $s'$, and use the shorthand

$$\mathbb{KL}_{s,s'}^{R,a,x} = \text{KL}(\mu_{a,x,s} \| \mu_{a,x,s'}) \ ,$$

for the KL-divergence between rewards under states $s, s'$. Our bounds and algorithms use a state $s$ as reference point for the set of alternative states $s'$ with different optimal arms,

$$\text{Alt}_x(s) := \{s' : a^*_{x,s'} \neq a^*_{x,s}\} .$$

We can now derive the following result.

**Lemma 1.** *Given a problem instance with latent state $s$ and observed context $x$, any $\delta$-PAC algorithm $\phi$ must satisfy for any alternative state $s' \in \text{Alt}_x(s)$,*

$$\sum_a \mathbb{E}_\phi[N_a \mid x, s]\mathbb{KL}^{R,a,x}_{s,s'} + \rho(x; s, s') \geq \mathbf{kl}(\delta||1 - \delta), \tag{2}$$

*where $N_a$ is the number of plays of arm $a$ drawn under $\phi$ and $\mathbf{kl}(\delta||1 - \delta)$ is the KL-divergence between two Bernoulli random variables with parameters $\delta$ and $1 - \delta$.*

**Proof summary.** *The proof follows the argument of the original Lemma in Kaufmann et al. (2016). We start from the KL-divergence between the distribution of histories $H$, under $s$ and $s'$ and expand this using the chain-rule of the KL-divergence. We then apply the information-processing inequality to lower bound this by $\mathbf{kl}(\delta||1 - \delta)$. The difference from Kaufmann et al. (2016) is that we get an additive term which depends on the context distribution under different latent models. For a full proof, see Appendix A.1.*

From lemma 1, we can derive a lower bound on the expected stopping time. Here, we assume that the optimal arm is unique for each state-context pair $(s, x)$, that is, $\text{Alt}_x(s) = \mathcal{S} \setminus \{s\}$. This assumption is *not* necessary to run our proposed algorithms.

**Proposition 1.** *For any $\delta$-PAC learner $\phi$ with $\delta \in (0, 1/2)$ and any latent state $s$ and context $x$, the expected stopping time satisfies*

$$\mathbb{E}_\phi[\tau \mid s, x] \geq \frac{1}{C^*_\delta(s, x)}\mathbf{kl}(\delta||1 - \delta)$$

*where $1/C^*_\delta(s, x) = \sum_a \gamma^*_{x,a}(s)$ with $\gamma^*_{x,a}(s)$ the minimizers of the following linear program,*

$$\underset{\gamma_{x,a} \geq 0}{\text{minimize}} \sum_a \gamma_{x,a} \tag{3}$$

$$\text{subject to } \sum_a \gamma_{x,a}\mathbb{KL}^{R,a,x}_{s,s'} + \frac{\rho(x; s, s')}{\mathbf{kl}(\delta||1 - \delta)} \geq 1, \ \forall s' \in \text{Alt}_x(s)$$

**Proof summary.** *By lemma 1, we have a constraint on the sum of the expected number of times each arm is played by any $\delta$-PAC algorithm $\phi$. By dividing each side of equation 2 by $\mathbf{kl}(\delta||1 - \delta)$ and minimizing the the stopping time under the resulting constraint, we obtain the linear program (LP) in equation 3. For a new bandit instance, $x$ is observed before search begins. Thus, given a model $\mathcal{M}_\theta$, the only unknowns in equation 3 are $\gamma_{x,a}$. As we have a finite set of latent states $s$ , we can construct a finite set of linear constraints and solve for the minimal stopping time. A full proof is given in appendix A.1.*

**Remark 1.** *As a sanity check, we verify that the contextual information makes the pure-exploration problem fundamentally easier. Indeed, when an observation $x$ clearly separates the true latent state $s$ from $s'$, $\rho$ increases, the constraint in equation 3 is satisfied by a larger set of parameters $\gamma_{x,a}$, and the lower bound attains a smaller value. However, as we require increasing certainty and $\delta \to 0$, the influence from contextual information $X$ on $C^*_\delta(s, x)$ vanishes. This is expected since we don't collect more information through $x$ as our requirement on certainty increases—it remains constant.*

As a consequence of proposition 1, we can obtain a bound for the population (marginal) search time. If we assume that $\frac{1}{C^*_\delta} = \mathbb{E}_{X,S}[\sum_a \gamma^*_{x,a}(s)]$ exists, with $\gamma^*_{x,a}$ the minimizers as in proposition 1, we have

$$\mathbb{E}_{\phi,X,S}[\tau] \geq \frac{1}{C^*_\delta}\mathbf{kl}(\delta||1 - \delta)$$

The lower bound indicates that the optimal worst-case solution to equation 1 is limited by the hardest-to-separate states $s, s'$. We make use of this insight next to develop algorithms.

---

**Algorithm 1** LLPT Explorer and Divergence Explorer

     **Input** $\delta, T, \mathcal{S}, K, \mathcal{M}_\theta$
     **Output** $\tau, \hat{i}_\tau$

  1: **Observe** $h_1 = (x)$
  2: **if** LLPT Explorer **then**
  3:      Compute $w_{x,a}^*(s)$ for all $a, s$                                       $\triangleright$ See equation 4, equation 3
  4: **end if**
  5:
  6: **while** $Z_t < 1 - \delta$ and $t < T$ **do**
  7:      **if** LLPT Explorer **then**
  8:          $s_t = \arg\max_{s \in \mathcal{S}} p(s|h_t)$
  9:          $a_{t+1} = \arg\max_{a \in [K]} \;\; t \cdot w_{x,a}^*(s_t) - N_{a_t}(t)$
10:      **else if** Divergence Explorer **then**
11:          $s_t \sim p(s|h_t)$
12:          $f_t(a) = \sum_{s'} p_\theta(s'|h_t) \mathrm{KL}(\mu_{a,x,s_t} \,\|\, \mu_{a,x,s'})$
13:          $a_{t+1} = \arg\max_{a \in [K]} f_t(a)$
14:      **end if**
15:      **Choose** $a_{t+1}$, and **Observe** $r_{t+1}$
16:      **Update** $h_t = h_{t-1} \cup (a_{t+1}, r_{t+1})$
17:      **Update** $N_{a_{t+1}}(t) \leftarrow N_{a_{t+1}}(t) + 1$
18:
19:      **Update** $\hat{s}_t = \arg\max_{s \in \mathcal{S}} p_\theta(s \mid h_t)$
20:      **Update** $\hat{a}_t = \arg\max_{a \in [K]} \mu_{a,x,\hat{s}_t}$
21:      **Update** $Z_t = \sum_s p_\theta(s|h_t) \mathbb{1}[\hat{a}_t = a_{x,s}^*]$
22: **end while**
23:
24: **Return** $\hat{a}_t$

---

## 4 Algorithms

We present two best-arm identification strategies, each comprising a sampling rule for selecting arms $A_t$, a stopping rule for determining $\tau$, and a recommendation rule for selecting $\hat{a}_\tau$. Both algorithms, defined in Algorithm 1, are given access to an *already estimated* latent variable model $\mathcal{M}_\theta$ including all reward means $\mu_{a,x,s} \;\forall\; s \in S, a \in A$ given a context $x$ and differ only in their sampling rules; the stopping and recommendation rules are equivalent. Either algorithm starts by observing the random context $X$, and proceeds from there.

### 4.1 Sampling rule 1: Latent LP-based Track and Stop (LLPT) explorer

Our first sampling rule is based on the Track-and-Stop method (Garivier and Kaufmann, 2016), where arm allocations are determined by tracking proportions $w^*$, obtained by solving the lower bound optimization problem in equation 3. Since we have finite sets of states and actions, and $x$ is observed at the start of the search, we can compute $\gamma_{x,a}^*(s)$ for all $s \in \mathcal{S}, a \in \mathcal{A}$ directly. Then, we define playing proportions $w_x^*(s)$, for each possible state $s \in \mathcal{S}$, as

$$w_{x,a}^*(s) = \gamma_{x,a}^*(s) / (\sum_a \gamma_{x,a}^*(s)) \;. \tag{4}$$

At each time step $t$, the algorithm picks a latent state $s_t = \arg\max_s p(s|h_t)$ from the (known) posterior given the current history $h_t$, and plays the arm which most closely tracks $w_{x,a}^*(s_t)$. Let $N_a(t)$ be the number of times arm $a$ has been played up until and including $t$. Then, the *LLPT Explorer* sampling rule is defined by

$$A_{t+1} = \arg\max_{a \in [k]} \;\; t \cdot w_{x,a}^*(s_t) - N_a(t) \;.$$

The LLPT Explorer aims to play the minimum total number of trials using arms which distinguish latent states the most, as given by the KL term in the constraint of equation 3. It aims only to distinguish latent states with different optimal arms, as the goal is to identify the best action, not the state.

## 4.2 Sampling rule 2: Divergence explorer

The LLPT Explorer plays according to the optimal proportions for the worst-case alternative state given the current estimate. This is because the constraint in equation 3 will be hardest to satisfy (require largest $\gamma_{x,a}$) for states $s'$ which are the most similar to $s$. A drawback of this idea is that it ignores the likelihood of said alternative state under the posterior. If there is strong evidence that $s'$ is unlikely to be the true state, collecting more evidence to rule it out may be suboptimal. In the extreme case, a state $s'$ with posterior probability $p(S = s' \mid h_t) \approx 0$ may still (unnecessarily) inform the sampling rule for the LLPT Explorer.

As an alternative, we define the *Divergence Explorer* sampling rule. This algorithm aims to play arms according to how much information is gained by playing an arm *in expectation* given the current posterior probability of states in $\text{Alt}_x(s)$. At each time $t$, a latent state $s_t \sim P_t(s|h_t)$ is sampled as reference. Then, the sampling rule uses the expected divergence between $s_t$ and alternative states $s'_t$,

$$f_t(s_t, a) = \sum_{s'_t \in \text{Alt}_x(s_t)} P(s'_t|h_t)\text{KL}(\mu_{a,x,s_t} \| \mu_{a,x,s'_t}) \ .$$

The arm $A_{t+1} = \arg\max_{a \in \mathcal{A}} f_t(s_t, a)$ is played next.

Because $\text{KL}(\mu_{a,x,s_t} \| \mu_{a,x,s'_t})$ measures the information distance between the reward distribution of arm $a$ under the two latent models $s_t$ and $s'_t$, $f_t(s_t, a)$ does a one-to-many test assuming $s_t$ is the true model and $s'_t$ is another latent model with probability $P(s'_t|h_t)$.

## 4.3 Recommendation rule

Both algorithms recommend the best arm in the state most believed to be correct in a given instance, so the recommendation rule is $\hat{a}_\tau = \arg\max_{a \in \mathcal{A}} \ \mu_{a,x,\hat{s}_\tau}$ where $\hat{s}_\tau$ is the most probable state under the posterior, as defined in Algorithm 1.

## 4.4 Stopping rule

It is natural to stop search at $t$ when we are confident enough that the recommended arm $\hat{a}_t$ is optimal under the posterior over latent states. Since we assume to have access to the full posterior over $S$, we can use the simple stopping rule

$$\tau := \min_t\{t : Z_t \geq 1 - \delta\} \quad \text{where} \quad Z_t = \sum_s P(s|h_t)\mathbb{1}[\hat{a}_t = a^*_{x,s}] \tag{5}$$

and the threshold $1 - \delta$ is the desired confidence level. Whenever this rule is satisfied, so is Chernoff's stopping rule based on a threshold $\log(\frac{1-\delta}{\delta})$ on the log-likelihood ratio between states, as used by Garivier and Kaufmann (2016). See the proof of proposition 2 in appendix A.2 for a derivation.

In many applications, it us sufficient to identify a action which is $\epsilon$-optimal with respect to the best possible action in the true latent state. We can accommodate this in our algorithm by redefining the set of alternative states $s'$ to include only those for which the optimal arm in $s$ is more than $\epsilon$ worse than the optimal arm in $s'$,

$$\text{Alt}_x(s) := \{s' : \mu_{a^*_{x,s},x,s'} < \mu^*_{x,s'} - \epsilon\} \ .$$

This change involves only a minor modification to the stopping criterion in equation 5 and could also be used in the Divergence explorer sampling rule.

# 5 Upper bound on the expected stopping time of LLPT explorer

Next, we show that the lower bound derived in Section 3 is matched by an upper bound on the stopping time for the LLPT Explorer algorithm in the high-confidence limit, $\delta \to 0$. Similar to the lower bound, we make

the simplifying assumption that each latent state has a unique optimal arm, shared with no other states, $\text{Alt}_x(s) = \mathcal{S} \setminus \{s\}$. As a consequence, finding the optimal arm equates to finding the true underlying state. We have the following result.

**Proposition 2.** *Let $\tau$ be the stopping time of LLPT Explorer $\phi$, as defined in Algorithm 1. With $s$ the true state and $C^*(s, x)$ the optimum in equation 3 with the $\rho$-term removed, there is a constant $\alpha > 0$ such that*

$$\lim_{\delta \to 0} \frac{\mathbb{E}_\phi[\tau \mid s, x]}{\log(1/\delta)} \leq \frac{\alpha}{C^*(s, x)} \ . \tag{6}$$

**Proof summary.** *The proof combines and expands arguments from Garivier and Kaufmann (2016) and Chernoff (1959) to show that after sufficiently many samples, a) the true latent state is identified, b) the tracked proportions are near optimal for the identified state, c) the probability that the stopping criterion is not satisfied decays exponentially quickly. As a result, the expected stopping time can be bounded using concentration arguments. For a proof, see appendix A.2.* ∎

As stated, proposition 2 applies to the LLPT Explorer, as defined in Algorithm 1, in which the MAP state $\hat{s}_t$ is used for tracking. We have also implemented a slight variation of LLPT with a sampled state $\hat{s}_t \sim p(s|h_t)$ and found that the latter worked slightly better empirically. We report only results for the version in Algorithm 1.

Similarly to the lower bound, we obtain an upper bound on the population search time by taking the expectation of equation 6 with respect to $S$ and $X$.

**Remark 2.** *Comparing the result in equation 6 to bounds for pure-exploration without latent variable models; e.g., Russo (2016); Garivier and Kaufmann (2016), superficially, they appear very similar. However, the critical quantity in the classical setting is the smallest separation of reward means for alternative, free vectors of arm parameters. Here, the equivalent quantity is the set of parameters of the discrete latent states, which is generally much smaller than the set of free parameters, leading to a tighter bound.*

*More precisely, the sample complexity term $C^*(s, x)$ shrinks when we have knowledge of the latent state structure because the set of plausible alternative parameters $\text{Alt}_x(s)$ is smaller compared to the case with no structure in, for example, Garivier and Kaufmann (2016). In our case, $\text{Alt}_x(s)$ comprises a finite set of parameters, whereas the case where parameters are estimated online without latent structure corresponds to an infinite set of alternative parameters. As a result, the worst-case (supremum) over alternative parameter sets shrinks, as do the lower and upper bounds on the stopping time.*

# 6 Experimental study

We evaluate our proposed algorithms in a series of experiments, comparing them to baseline algorithms for fixed-confidence pure exploration.

## 6.1 Baseline algorithms

Previous work incorporating latent states in pure exploration was not available at the time of writing, so to get comparable baselines, we adapted the Top-Two Thompson Sampling (TTTS) rule (Russo, 2016) to compare to our algorithms.

**Top-Two Thompson Sampling (TTTS)**  TTTS operates with the goal of estimating parameters $\Pi_t$ (e.g., mean vectors of arms with Gaussian distribution) that yield the best arm for a given confidence level $1 - \delta$. It proceeds as follows; at each time step $t$ either; (i) with probability $p$, sample a parameter vector $\theta_t \sim \Pi_t$ and play the arm $a_t^{(1)} = \arg\max_{a \in \mathcal{A}} \theta_t$ or (ii) with probability $1 - p$ resample $\theta_t' \sim \Pi_t$ until it gets and subsequently plays arm $a_t^{(2)} \neq a_t^{(1)}$. We implemented the T3C (Shang et al., 2020) variant of TTTS which finds $a_t^{(2)} \neq a_t^{(1)}$ faster. TTTS does not make use of a latent variable model.

**TTTS-Latent Explorer**  This is an adaptation of TTTS to our setting where, instead of estimating arm parameters, the goal is purely to identify the latent state. It does not account for the case where there is a shared optimal arm over different states which is accounted for in the LLPT and Divergence Explorer.

At each time step $t$, the sampling rule samples a latent state $s_t^{(1)} \sim P_t(s|h_t)$ and either (i) with a Bernoulli parameter $p$ evaluates the latent state, $s_t = s_t^{(1)}$ or (ii) with a Bernoulli parameter $1 - p$ resamples $P_t(s|h_t)$ until it gets a latent state $s_t = s_t^{(2)} \neq s_t^{(1)}$. It then plays the arm $A_t = \arg\max_{a \in \mathcal{A}} \mu_{a,x,s_t}$.

**Greedy Explorer** This is a naïve sampling rule which plays the reward-optimal arm in a state sampled from the current posterior, akin to TTTS-Latent but without the re-sampling step. At each time $t$, it picks $s_t = \arg\max_s p(s|h_t)$ and then plays the locally reward-maximizing arm $A_t = \arg\max_{a \in \mathcal{A}} \mu_{a,x,s_t}$. It is naïve in the sense that it only considers the rewards from a state, but this is not always informative for distinguishing alternative states. It also corresponds to standard Thompson Sampling (Thompson, 1933) which has been shown to perform poorly for pure exploration tasks, hence the motivation for TTTS.

## 6.2 Experimental environment

As treatment personalization task, we use the Alzheimer's Disease Causal estimation Benchmark (ADCB) environment (Kinyanjui and Johansson, 2022). In this environment, simulated subjects go through cognitive decline, eventually progressing into Alzheimer's disease. Outcomes $Y_t$ represent their cognitive abilities and treatments $A_t$ are symptomatic, affecting only immediate outcomes. Both treatment responses and an initial 33-dimensional observed context $X \in \mathbb{R}^d$, are affected by a latent state $S$, representing the disease subtype.

In the ADCB environment, the number of actions is $K = 8$ and the number of latent states, $S = 6$. The outcome $Y_t$ at time $t$ is generated as $Y_t(A, X, S) := \Phi(X, S) + \Delta(A_t, S) + \xi$, where $\xi \sim \mathcal{N}(0, \sigma^2)$ and $\Phi$ is an non-linear function fit to real data to model the cognitive function of subjects when not treated. For the environment we are using, $\Phi$ is a Random Forest Regressor fit to observed outcomes of untreated patients. $\Delta$ is a function that is defined to moderate the heterogeneity of simulated treatment effects over the latent dimensions. Here, $\Delta := \upsilon \mathbb{1}_S + \mathbb{1}_S(\eta \upsilon \beta^T)$ where $\upsilon \in \mathbb{R}^K$ is the average treatment effect of the treatments, $\eta > 0$ is a heterogoneity scaling parameter, and $\beta \in \mathbb{R}^{K \times S}$ is a factor matrix whose rows sum to 0.

We define two alternative reward settings (see below), both with Gaussian rewards, based on the ADCB outcomes of treatments, $Y$. We give algorithms which make use of latent variables perfect knowledge of the true latent variable model, as defined by the simulator. Hence, for each context $x \in \mathbb{R}^d$, latent state $s \in [S]$ and action $a \in [K]$, the corresponding posterior $p(s \mid h_t)$ and reward means, $\mu_{a,x,s}$ are known.

**Reward setting 1: Non-contextual rewards** Here, for each latent state we define the reward $R := -(Y(A, X, S) - Y(0, X, S))$. From the definition of the outcome $Y$ above, this removes the effect of context from the reward, by cancelling $\Phi(X, s)$, and takes us closer to a typical best arm identification setting with additional latent state structure, where the structure is given by $\Delta$. In appendix B.1, Figure 5(a) shows the structure of the mean rewards $\mu_{a,x,s}$ under the different latent states $s \in \mathcal{S}$, $a \in K$ for this setting.

**Reward setting 2: Contextual rewards** Here, we define the reward $R := -Y(A, X, S)$, thus preserving the effect of context in the reward. As seen from appendix B.1, Figure 5(b), which is an example of the mean rewards structure $\mu_{a,x,s}$ $s \in \mathcal{S}$, $a \in K$ for some given context $x$, the reward structure stays the same as in the previous setting, but the scale is shifted depending on the context. The similarity is a property of the environment. The results presented in the results section below are for this setting, and those of setting 1 above are appended in the supplementary materials.

**Repeated experiments** Each experiment proceeds as follows; A new patient is sampled from the environment (sampled patients have potentially different latent states and contexts). The algorithms do not observe the latent state and they proceed as described in Section 4 and Section 6.1. For a run, all algorithms are provided with the same context. All results are presented for 100 different patients and averages are computed for the different quantities compared. Errorbars represent the standard deviation across patients.

**Evaluation metrics** We compare empirical estimates of the expected stopping time $\mathbb{E}[\tau]$, convergence of the posterior probability $p(\hat{s}_t \mid h_t)$ with $t$, and the average correctness level, $\mathbb{E}[\mathbb{1}[\hat{a}_\tau = a^*]]$, of the different algorithms for i) different levels of confidence $\delta \in (0, 1/2)$ under a fixed noise level $\sigma > 0$ and ii) different levels

of noise $\sigma$ for a fixed $\delta$. Results for correctness are presented in Figure 6 in the Appendix, and correspond closely with the parameter $\delta$.

### 6.3 Results

In Figure 2, we see an example of the drastic effect that incorporating latent structure can have on the stopping time of pure-exploration algorithms. All latent-variable methods outperform the non-latent baseline TTTS by a substantial margin.

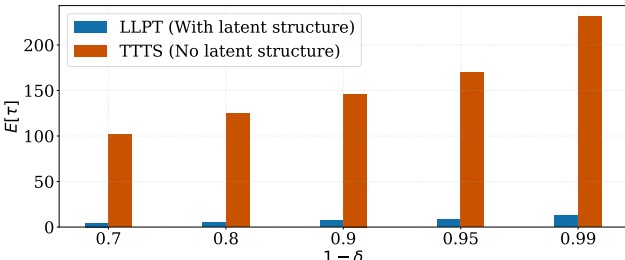

Figure 2: Using latent state structural information significantly reduces the expected number of trials $\mathbb{E}[\tau]$ required to identify an optimal treatment with confidence at least $1 - \delta$ in a simulator of Alzheimer's disease progression.

Moreover, in Figure 3a, we see that, even for the worst-case instances, the LLPT algorithm is faster than the average for standard TTTS observed in Figure 2. This supports our hypothesis that exploiting latent structure between instances (patients), which could be estimated from historical data, contexts, is useful to design sample-efficient pure-exploration algorithms.

In the graph of latent state posterior convergence, Figure 3b, we see that LLPT Explorer and Divergence Explorer converge quicker in their belief of the inferred latent state. We also observe less variance across bandit instances (shaded area) compared to the Greedy and TTTS-Latent baselines. The implication for this is that these algorithms stop exploration earlier thus attaining our goals outlined in Section 2.

In Figure 3c, we study the average stopping time, $\hat{\mathbb{E}}[\tau]$ for all algorithms with access to the same latent variable model, under changing certainty level $1 - \delta$. LLPT Explorer and Divergence Explorer are consistently more efficient than baselines, demonstrating benefit of the insights derived from the lower bound in proposition 1. The difference is especially pronounced in the high-certainty regime, $\delta \approx 0$, which is the regime that would be ideal for safety-critical healthcare applications. Interestingly, we find that the Divergence Explorer performs consistently better than the LLPT Explorer and its average stopping time approaches the lower bound as $\delta \to 0$. We believe this is due to selecting actions based on comparison with alternative states on average under the current posterior, rather than the worst-case alternative state - some latent states are ruled out by the posterior and no longer affect the action selection of the divergence explorer.

Studying our algorithms with respect to noise in the rewards, Figure 3d, shows that our proposed methods are also more robust to noise compared to the baseline algorithms. At $\sigma = 10$, which is comparable to the marginal standard deviation of rewards due to $X$ and $S$, we see that our algorithms perform better. We also observe that they are also more robust to over- and under-estimation of the noise level in the rewards as shown by $\mathbb{E}[\tau]$ at other noise levels.

## 7 Related work

The problem of finding optimal decisions under uncertainty has a long history (Thompson, 1933; Chernoff, 1959; Gittens and Dempster, 1979; Jennison et al., 1982; Lai and Robbins, 1985; Glynn and Juneja, 2004) and has recently been studied as a pure exploration problem in the multi-armed bandit framework under

---

[2]The small discrepancy seen in the case where $\sigma = 1$ is due to the exclusion of the $\rho$ term in the computed lower bound.

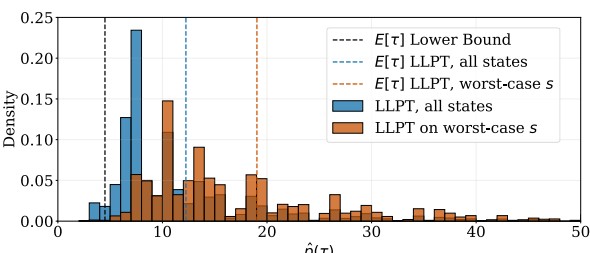

(a) Density of stopping times under LLPT(ours) showing worst-case latent state ($\delta = 0.01$, Number of patients, $N = 10,000$). The variance of the stopping time under all the latent states is reasonably low. The higher stopping times can be attributed to the worst-case latent states, though they are still reasonably low.

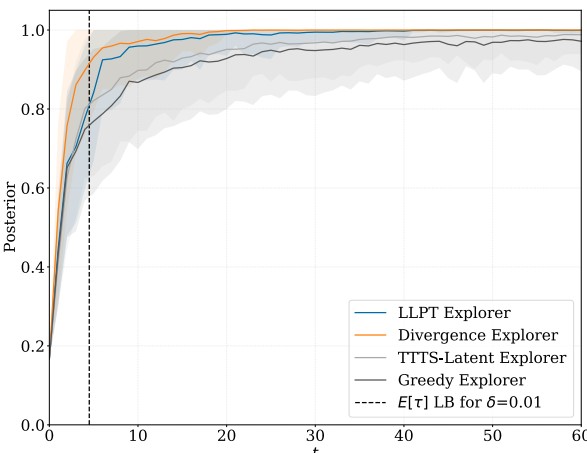

(b) Comparison of posterior convergence of the different algorithms [$\delta = 0.01$, Number of patients, $N = 100$]. The posteriors for our algorithms, LLPT Explorer and Divergence Explorer, concentrate more quickly.

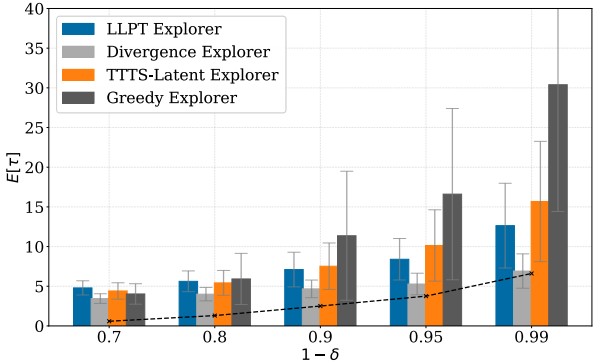

(c) Comparison of stopping time vs confidence $(1 - \delta)$ for the algorithms. Our algorithms, LLPT Explorer and Divergence Explorer, have stopping times that are consistently lower. The dashed line shows the lower bound from Proposition 1.

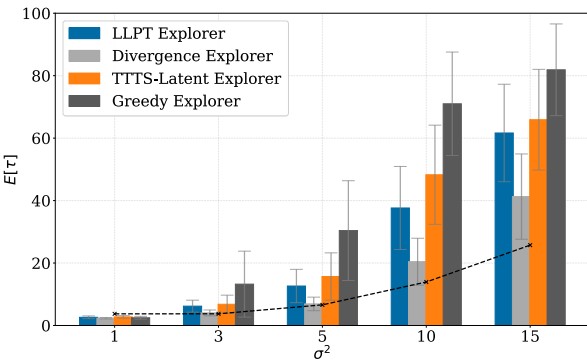

(d) Comparison of stopping time vs noise for the algorithms Our algorithms, LLPT Explorer and Divergence Explorer, are consistently more robust to noisy rewards compared to the baselines. The dashed line shows the lower bound from Proposition 1.[2]

Figure 3: Selected results from our experimental study.

various assumptions(Even-Dar et al., 2006; Bubeck et al., 2009; Jamieson et al., 2013; Kaufmann et al., 2016; Garivier and Kaufmann, 2016; Jedra and Proutiere, 2020; Wang et al., 2021; Agrawal et al., 2021; Tirinzoni and Degenne, 2022).

The work of Garivier and Kaufmann (2016) is the first to introduce an optimal algorithm, Track and Stop, in the fixed confidence setting for classical multi-armed bandits and our LLPT Explorer takes inspiration from their algorithm, adapting it to the latent bandit setting. Russo (2016) introduces a class of top-two sampling strategies for the pure-exploration problem, which we here use as baselines. These top-two algorithms were originally analyzed using a different performance measure but have recently been theoretically analyzed in the fixed-confidence setting by Jourdan et al. (2022). Our work is also related to (Maillard and Mannor, 2014; Zhou and Brunskill, 2016; Hong et al., 2020a;b), who study regret minimization in latent bandits, in contrast to our work which studies the pure-exploration problem in latent bandits.

Kato and Ariu (2021) studied pure exploration in contextual bandits, where a new context is observed at each time point, and found that contextual information improves the speed at which the average treatment

effects (Imbens and Rubin, 2015) of actions across contexts can be estimated. Our problem is related to this setting but differs in that we see only a single context $x$ per bandit instance, and are interested in the effects of actions for this specific $x$, not on average. Håkansson et al. (2020) studied fast search for near-optimal treatments, based on a model learned from historical trajectories, but did not consider online learning. In their setting, an optimal search strategy can be found by solving a dynamic programming problem in an estimated discrete state space. This is not feasible here due to the high dimensionality of our history, $H$.

## 8 Discussion & conclusion

In this work we have studied the problem of finding the optimal arm in latent bandits using as few trials as possible. We have empirically and theoretically shown that our proposed algorithms are able to leverage the latent structure in a near-optimal way to substantially reduce the expected stopping time compared to available baselines. Our empirical evaluation in a simulator of Alzheimer's disease derived from real-world data, demonstrated that our algorithms are able to find the optimal treatment in just a few trials.

Our analysis is limited to the case in which the latent variable model is given and exact. When forced to estimate the model from historical data, sensitivity to misspecification or misestimation becomes a concern. Hong et al. (2020a) analyzed latent bandits in regret minimization when the reward model is misspecified but the resulting bound suffers linear regret scaled by the error, and Hong et al. (2022) provided an improved sub-linear regret bound for this with additional assumptions on the reward structure. In the pure-exploration setting, recovering quickly from misspecification is even more critical since the time scale is shorter. We conjecture that an informative guarantee in the misspecified case will similarly require additional assumptions on the reward structure or additional sources of data. We believe the setting where a learner needs to recover the true model up to some pre-specified precision is an interesting direction for future work. Another useful generalization would be to go beyond the analysis of expected rewards. In high-stakes applications, it is desirable to manage also the risk of worst-case low-probability events, see e.g., Tamkin et al. (2019). This would further increase the suitability of our approach for the medical domain.

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
