# OpenReview forum: "Fast Treatment Personalization with Latent Bandits in Fixed-Confidence Pure Exploration"
_TMLR — Accepted by TMLR_

### Review · Reviewer_nfdE · 2023-03-10

**Summary Of Contributions:**

This paper considered latent bandits in fixed-confidence pure exploration setting, with application to personalized treatments for patients. The authors considered a contextual bandit setting, where context information is not sufficient to identify the reward and optimal action. Latent state is considered as unobserved information that determines the reward jointly with context. The authors proposed two methods, Latent LP-based Track and Stop (LLPT) Explorer and Divergence Explorer. The authors also proved a lower bound on stopping time of latent bandits that matched the upper bound of LLPT. Experiments on simulation of Alzheimer’s disease validated the advantage of the proposed methods.


**Audience:**

Yes

**Claims And Evidence:**

Yes

**Requested Changes:**

See weaknesses and questions for requested changes.


**Strengths And Weaknesses:**


Strengths:
1. The problem of latent bandits in pure exploration is practical and well-motivated.

2. LLPT Explorer solves minimal stopping time that distinguishes latent state by the LP, which is an intuitive idea. Theoretical analysis of the lower bound and the matching upper bound based on LP is a good contribution of the paper. Compared to the lower bound and the upper bound of Track-and-Stop strategy in Garivier and Kaufmann (2016), here the bounds of latent bandits depend on the identification of latent state that is described by the LP.

3. Extensive experiments showed the proposed two methods consistently outperformed baselines under different confidence and noise level.

Weaknesses and questions:

1. It is unclear why Divergence Explorer consistently performed better than LLPT Explorer especially since Divergence Explorer does not have a stopping time upper bound analysis. The authors are encouraged to at least mention the challenge of analyzing of Divergence Explorer (e.g., maybe a worst-case guarantee cannot be analyzed for Divergence Explorer). If such worst-case bound does not exist, is it possible to show that there exist some hard instances that LLPT performs similarly or better than Divergence in experiments?

2. TTTS-Latent Explorer performed similarly to LLPT Explorer in experiments. Can we adapt the theoretical guarantee of TTTS and be able to obtain an upper bound of TTTS-Latent Explorer for latent bandits? Why LLPT has a similar performance as TTTS-Latent? Is it possible to show that the performance gap could be larger with different latent state settings in experiments? It would be better to also explain the similarity/gap from a theoretical perspective.

---

> ### Author Response · Authors · 2023-03-22
> **Response to Reviewer nfdE**
>
> **Re: Comparison between Divergence Explorer and LLPT**
>
> As discussed briefly in the manuscript at the end of page 10, we believe that the advantage of the Divergence Explorer (DE) comes down to the posterior ruling out some of the latent states during exploration. As a result, these do no longer affect the action selection of DE (see ln.9 in Algorithm 1). In comparison, the LLPT explorer will continue to track weights which are influenced by states which may be very unlikely under the observed rewards.
>
> **Re: Analyzing  divergence explorer**
>
> Divergence explorer does not track optimal proportions like LLPT, so we cannot rely on proof techniques from Track and Stop to analyze it. We are not aware of an existing proof technique that applies to this setting, and we consider the analysis of the Divergence Explorer an interesting challenge for future work. We have yet to find a concrete example where LLPT performs similarly or better than Divergence explorer in experiments, but this is a good question that we'll look into as we do our revision.
>
>
> **Re: TTTS-Latent compared to LLPT**
>
> It is incorrect that TTTS-Latent performed similarly to LLPT explorer, because at higher confidence levels it performs consistently worse. However, it's a good idea to look into adapting proof techniques of TTTS to the latent bandits setting, and this could be a direction for future work.

---

> > ### Comment · Reviewer_nfdE · 2023-04-07
> > **Response to authors**
> >
> > Thank you for the clarification and revision of the paper.

---

### Review · Reviewer_F7AY · 2023-03-11

**Summary Of Contributions:**

This paper studies the fixed confidence pure exploration problem for a $K$-armed bandit with a latent state. There are $M$ possible states of the bandit, and the identity of the optimal arm for each of these $M$ states is known (together with the distribution of arm-rewards in each possible state), however, the state itself is unknown a priori. The problem thus reduces to that of latent state estimation using rewards as observations.

A sample complexity lower bound is derived for this problem; the authors also propose two algorithms one of which is shown to attain the lower bound up to constant multiplicative factors.

**Audience:**

Yes

**Broader Impact Concerns:**

None.

**Claims And Evidence:**

Yes

**Requested Changes:**

Few typos:

1. ALG1 line 13 should be: $a_{t+1} = \arg\max_{a\in[K]} t w_{x,a}^\ast(s_t) - N_a(t)$.

2. The constant in equation (6) should be $C_0^\ast(s,x)$, right? If this is not the case, then the bounds do not match, correct?

**Strengths And Weaknesses:**

The problem is well-motivated, and the paper well-written.

Questions:

1. How rich is the context set? The formulation would make sense only if the set of possible contexts is infinite. If it is finite, the optimization problem should be solved for each possible context. Essentially, contexts do not add to the complexity of the problem in this case. In light of this, I have a question from the first paragraph on page 3 (line 7) -- "To account for remaining individual variation between patients with the same X, we will assume....." It seems from this line that the context set X is assumed to be finite (otherwise patients with same X would occur with probability 0 in general). Could the authors please clarify upon this?

2. In continuation of the above, if X is indeed assumed finite, wouldn't it simplify exposition (without loss of any generality) if the context is completely removed from the model altogether? I think the paper fundamentally is about latent state estimation. Presence of context only adds unnecessary detail. The way I would motivate this setting (at least in my mind) would be to consider a classical 2-armed problem with a fixed confidence BAI objective. Where the problem would differ from traditional BAI would be in the presence of auxiliary information. For example, the bandit could have 2 states: one with $\mathcal{N}(0.5,1)$ and $\mathcal{N}(1,1)$ rewards for arms 1 and 2 respectively; and the other with reward distributions exchanged. The decision maker knows, for each possible state, the distribution of rewards for the two arms, but the state is latent, and therefore the task reduces to figuring out which of the two arms has distribution $\mathcal{N}(1,1)$. Would this example fit into the framework?

3. Continuing the above, I would like to see a more comprehensive discussion of how ex ante knowledge of the tuple of reward distributions for each possible state leads to a reduction in sample complexity (the $C_\delta$ factor).

---

> ### Author Response · Authors · 2023-03-22
> **Response to Reviewer F7AY**
>
> **Re: Finite context**
>
> We do not consider the context to be finite. The utility of the context is in computation of the likelihood $P(s| H_t)$ and this is agnostic of either finite or infinite context assuming that a good model of the likelihood is known. Reviewer F7AY is correct that the probability of observing two subjects with identical contexts is 0 in the infinite-context case. The sentence the reviewer refers to was meant to illustrate that *even if* two subjects with the same context were observed, they would not necessarily share the same optimal treatment, since they may have different latent states. We will clarify this in the revision.
>
> **Re: Identification of the latent state**
>
> Identifying the correct latent state is sufficient but not strictly necessary to solve our problem. For successful treatment, we are only interested to identify the optimal action. Therefore, it is not necessary to estimate the correct latent state, but the set of latent states that have the same optimal arm. Having a context is desirable as it helps  reduce the number of trials if it is informative of the underlying latent state $S$, and residual unexplained variation could be explained by trying different treatments. Moreover, in our intended applications, it is common that contextual information is available.
>
> **Re: The complexity term $C_\delta$**
>
> The sample complexity term $C_\delta$ shrinks when we have knowledge of the latent state structure because the set of plausible alternative parameters $Alt(s)$ is smaller compared to the case with no structure in, for example, Garivier \& Kaufmann (2016). Concretely, in our case, $Alt(s)$ comprises a finite set of parameters, whereas the case where parameters are estimated online without latent structure corresponds to an infinite set of alternative parameters. As a result, the worst-case (supremum) over alternative parameter sets shrinks, as do the lower and upper bounds on the stopping time.
>
> **References**
>
> [1] Garivier and Kaufmann. Optimal best arm identification with fixed confidence. Annual Conference on Learning Theory. PMLR, 2016.

---

> > ### Comment · Reviewer_F7AY · 2023-04-06
> > **Reply to authors**
> >
> > Thank you for answering my questions! I hope some of these clarifications are incorporated in the revision. Good luck!
> >
> > Best regards,
> > Reviewer F7AY

---

### Review · Reviewer_f3zP · 2023-03-13

**Summary Of Contributions:**

The authors consider a best arm identification problem for contextual bandits in the fixed confidence setting, with a latent state. Namely, an agent observes the context $x$, takes an action $a$, and obtains a reward with mean $\mu_{a,x,s}$ where $s$ is a state unknown to the learner, and where the mapping $(a,x,s) \mapsto \mu_{a,x,s}$ is known to the learner. This is repeated until the learner has identified the best action maximizing $a \mapsto \mu_{a,x,s}$ with probability greater than $1-\delta$ where $\delta$ is a confidence threshold known to the learner. It is noted that both state $s$ and context $x$ are static, and do not change during the exploration process.

In fact the problem is similar to clustered bandit, where the notion equivalent to the unobserved state $s$ is the cluster to which the bandit problem belongs.

The authors prove an information theoretic lower bound on the amount of samples required in order to actually identify the best arm reliably, as well as algorithms matching those bounds. Their approach is mainly an adaptation of the techniques developped in the stateless context by Kauffman and Garivier. The main originality from the mathematical side is that the set of parameters $\mu_{a,x,s}$ is not continuous like in the classical context, rather it is discrete of size $|S||X||A|$ which allows much faster learning.

**Audience:**

Yes

**Broader Impact Concerns:**

Not applicable.

**Claims And Evidence:**

Yes

**Requested Changes:**

- Give motivation for choosing best arm identification over regret maximization in the context of treating diseases.
- Show practically relevant numerical experiments where greedy strategies fail so that the problem is an actual bandit problem or numerical experiments where the greedy strategies are vastly outperformed by using bandit algorithms.


**Strengths And Weaknesses:**

Strengths:
- The paper is very well written
- The results seem novel, and the medical application well explored, both in terms of modelling as well as in terms of numerical experiments. The authors provide a complete set of results: lower bound, algorithms with a matching upper bound, and numerical experiments.

Weaknesses
- The main motivation for the model is the treatment of disease, which the authors use to motivate the model of best arm identification. Now, while this probably depends on practical considerations, I do not see why, in the treatment of a disease for an individual patient, one would like to perform best arm identification. Rather an individual patient cares much more about being allocated good treatments, which  in that context would mean minimizing regret. Imagine that there are 3 treatments, where both 1 and 2 are very good (say within a factor +- epsilon of each other, and 3 is extremely bad. Performing best arm identification is mostly overkill here, as it would require to differentiate between 1 and 2. Rather, minimizing regret would be more than sufficient, as it would guarantee that the bad treatment is seldom used. Identifying the best treatment seems useful at the population level (to make recommendations to the general public once the best treatment has been identified), but at the individual level identifying the best treatment seems much less useful. Also, if the disease is chronic, then the allocation of treatment never stops, rather it is a lifelong process, so that stopping at a given time to output the best treatment (like in best arm identification) is not necessary.
- Some of the numerical results seem a bit too good to be true, in the sense that some of the figures give the impression that, when using the structure, one can sample 5 arms or less and still find the optimal arm with high confidence. This would suggest that the model is so highly structured that a simple greedy approach works just fine, and exploration is not that necessary (essentially playing any action $a$ gives so much information about the state $s$ that we quickly figure $s$ out and play the corresponding best action). And indeed, based on the error bars in Figure 3, it is not even clear that the proposed algorithms do much better than greedy exploration. It would be much more interesting to look at experiments where greedy strategies fail, so that the bandit setting is actually justified.
- (Minor) The names used for $x$ "context" and $s$ "state" are somehow counter intuitive since none of them change during the learning process

---

> ### Author Response · Authors · 2023-03-22
> **Response to Reviewer f3zP**
>
> **Re: Motivation for choosing best-arm identification over $\epsilon$-best-arm or regret minimization in the context of treating diseases**
>
> We agree that it is plausible to consider a regret minimization formulation for our setting, and that it is often sufficient to identify an $\epsilon$-optimal therapy (see next paragraph). In our motivating applications, e.g., the treatment of chronic disease such as rheumatoid arthritis, switching therapies has associated costs: every time a therapy is changed, the patient has to get used to the new drug and its potential side effects. Thus, it is desirable to minimize such switches, even if changes are to other near-optimal treatments. We therefore formulate treatment as fixed-confidence pure exploration in order to minimize exploration both in the treatment search phase while also ensuring that the algorithm commits to a good treatment after exploration. Regret minimization does not have this incentive. We will clarify this further in the revision.
>
> In our problem formulation, $\epsilon$-optimality could be accomodated by letting the $Alt(s)$ set be the latent states $s'$ whose rewards for the optimal arm in $s$ is more than $\epsilon$ away from the the optimal reward in $s'$. This set is known a priori. Solving our problem with this strategy would yield a treatment that is one of the $\epsilon$-optimal treatments instead of only the best treatment after exploration stops. Deciding $\epsilon$ would require domain knowledge of the range of treatment effect efficacy which might change with time, and the efficacy might also vary from patient to patient. We will comment on this generalization as well.
>
> **Re: Practically relevant experiments where greedy strategies fail so that the problem is an actual bandit problem or numerical experiments where the greedy strategies are vastly outperformed by using bandit algorithms.**
>
> In our experiments for $\delta=0.01$ (see Figure 3c), the greedy baseline has an average stopping time which is more than twice as long as for the LLPT explorer, with very high variance, and more than four times as long as Divergence Explorer. In our intended application, this is a vast difference since the stopping time represents the number of therapies tried by a patient.
>
> More generally, there are latent structures where the optimal arms in different latent states aren't the most informative to differentiate the latent states. In such structures, it would be better to play well-separating arms that aren't optimal for any of the latent states, see an example below.
>
> **Example**: Consider a setting with $s=3, k=4$ with mean parameters $\mu_{s=1} = \\{1.9, 2.4, 2.3, 2.0\\}, \mu_{s=2} = \\{1.2, 2.3, 2.5, 2.3\\}, \mu_{s=3} = \\{0.4, 2.0, 1.9, 2.5\\}$ and Gaussian-distributed noise with standard deviation $\sigma$. In this example, it would be more informative to play arm $1$ which is a well-separating arm to distinguish the latent states, although it is not the optimal arm for any of the states. However, greedy strategies would always play the other arms, since each of them is the optimal arm for one of these states. With a sufficiently high noise level $\sigma$, a high number of plays of these arms would be required to distinguish them compared to playing the informative arm 1.
>
> Finally, we thank reviewer f3zP for pointing out the connection to clustered bandits. We will include this in our related work for the revision.

---

### Review · Reviewer_k5Me · 2023-03-14

**Summary Of Contributions:**

This paper studies the problem of personalized treatment of patients. The problem is framed as a sequential as a sequential decision-making problem, while assuming that some historical data of a population of patients are available. The decision-making part is mathematically formulated as a latent MAB problem in the fixed-confidence pure exploration setting. The paper considers an intermediate setting where the latent structure and mean rewards are known, and the challenge remains in identifying the latent state (of the patient in question). The first contribution is an extension of the existing lower bound (for pure exploration) to this setting. Then, two algorithms are presented. The first one is LLPT, which is an adaptation of the Track-and-Stop algorithm for conventional MABs. The second one is Divergence Explorer. The two algorithms have the same recommendation and stopping rules but differ in the choice of sampling rules. LLPT is shown to achieve the presented lower bound in the high-confidence regime. Numerical experiments are performed to justify the superiority of the presented algorithms over the relevant baselines from the MAB literature.

**Audience:**

Yes

**Broader Impact Concerns:**

Not necessary.

**Claims And Evidence:**

Yes

**Requested Changes:**

See above.

**Strengths And Weaknesses:**

Formulating personalized treatment search as pure exploration in latent MAB sounds interesting. Latent MABs appear to be the most reasonable candidate in modeling personalized treatment as a sequential decision making. Establishing this link through a solid mathematical formulation makes a nice contribution. Another strong point is to include both lower bound and algorithms. Finally, I found the setting of the numerical experiments quite relevant to the studied problem.

The paper restricts to an intermediate setting where the latent structure is assumed “fully known”, an assumption which may render unrealistic in many practical situations. This assumption implies that the distribution of latent states, the distribution of contexts (conditioned on the latent state), and even conditional distribution of rewards are perfectly known –note that this is stronger than assuming known mean rewards. As a result, once the latent state is accurately identified, the optimal action is deduced. Although it is argued that the said distributions could be accurately estimated using state-of-the-art techniques, I personally found the assumption rather strong, thus limiting the contribution. I agree that some components of the latent structure could be accurately estimated. However, this may no longer be true for all the components, for example, reward distributions.

It is nice to have included a lower bound for this setting. However, because of the perfectly known latent structure, its derivation straightforwardly follows from that of classical pure exploration (e.g., in (Kaufmann et al., 2016)).

The opening part of the article is written well, and the choice of problem formulation is reasonably justified. However, the rest, which constitutes the bigger part of the article, is written carelessly and perhaps in rush, which is strange for a journal submission with strict no deadline. Even the reference section contains strange mistakes: For example, see (Chernoff, 1959), where the paper title is missing. Another example is (Gittins and Dempster, 1979). The related work section discusses most relevant paper that I am aware of on the MAB side. There has been some recent work on RL in latent MDPs. Perhaps discussing this line of work and contrasting them to the considered problem could strength the literature review.

Notation mistakes abound throughout the paper. For example, in one place there is $\gamma^*$ depends on $s$, whereas this dependence disappears elsewhere without being justified. Another example: both $w^*_x(s)$

and $w^*_{x,a}(s)$ are used (Section 4.1). I listed further at the of this section.

Also, there are inconsistencies between the appendix and the main text in terms of notations and definitions. The mean reward in the main text correctly depends on $a,x,s$ whereas in the appendix it is a function of $s,a$ ---see Eq. 8, Appendix A.1. Also, $\mathrm{Alt}(s)$ should be $\mathrm{Alt}_x(s)$. Also, the expected value of $N_a$ should be conditioned on $x,a$, and not $X$. A similar comment applies to Eq. 12. Although all these appear fixable in one-shot, they evidently indicate that the authors did not do a careful proofreading.


The LLPT Explorer sampling rule is the same as the one in the conventional MABs. From the presented sampling rules, it is evident that the known latent structure is instrumental to straightforwardly tailor the sampling rules of classical pure exploration to this setting. Overall, my biggest concern is the imposed assumptions makes the contribution very marginal. On the theoretical side, the contribution is comparable to some light technical exercise. Moreover, the current execution suggests that the paper can still benefit from a careful rewriting.


---- Further detailed comments:

- In the constraint of the optimization problem (1), one must have strict inequality in the argument of $P$, i.e., $\mu_{\hat a_\tau, x,s} < \mu^*_{x,s}$.

- The notation in the algorithm is not consistent with the text. For example, history is denoted by $h_t$, whereas $H_t$ is used to denote it in the text. As a matter of fact, there are many others.

- The presentation of TTTS appears confusing to me.

---- Minor comments:

- Proposition 1: “…. the minimizers of the following linear program $x$” => Is $x$ redundant?

- Section 5: sufficently => sufficiently

- p. 8: an non-linear => a non-linear

- Algorithm, throughout: $h$ => $H$

- Algorithm, line 13: $N_{a_t}$ => $N_{a_t}(t)$ ---to be consistent with line 17

- Algorithm, line 20: $[k]$ => $[K]$

- In the proof of Lemma 1, line 1: remove “for”.

- In Appendix A.2, $L_n(s,s’)$ should be $L_t(s,s’)$.

---

> ### Author Response · Authors · 2023-03-22
> **Response to Reviewer k5Me**
>
>
> **Re: Strong assumptions**
>
> Reviewer k5Me is correct that the assumption of knowing a perfect model $\mathcal{M}_\theta$ is strong, but it is a natural starting point as there is currently no existing analysis for this setting with pure exploration. Moreover, the assumption  matches that of previous work in latent bandits in the regret minimization setting, such as Maillard \& Mannor (2014) and Zhou \& Brunskill (2016) and Hong et al., (2020). We believe that giving guarantees under a weaker assumption is possible, at the cost of longer exploration since the likelihood will be less informative, and view this as an exciting direction for future work.
>
> **Re: Writing and related work**
>
> We thank Reviewer k5Me for pointing out mistakes in the writing. In our revision, we will correct any typos, reference issues, and  inconsistencies in our notation as well as improve the overall writing in the later sections of the manuscript. A clarification to note is that $h$, as used in the algorithm description, is an observation and $H$ is a random variable, as per convention.
>
> We also thank the reviewer for the suggestion of including a discussion of reinforcement learning in latent MDPs, and plan to add that to our first revision.
>
> **References**
>
> [1] Hong et al., Latent bandits revisited, Advances in Neural Information Processing Systems, 2020.
>
> [2] Maillard and Mannor, Latent bandits, International Conference on Machine Learning, 2014.
>
> [3] Li Zhou and Emma Brunskill. Latent contextual bandits and their application to personalized recommendations
> for new users, arXiv preprint, 2016.

---

### Author Response · Authors · 2023-03-22
**Responses to Reviewers**

We thank all reviewers for their thoughtful and constructive feedback. We've now added our preliminary responses to each reviewer as we work on the suggested revisions. Should the reviewers have additional questions or clarifications, we're happy to discuss more.

---

### Author Response · Authors · 2023-03-27
**Revision submitted**

Thank you, to all the reviewers and the action editors, once again. We have now submitted the revision according to the feedback received. Looking forward to hearing back.

---

### Author Response · Authors · 2023-04-28
**Camera ready version submitted**

We are happy to receive the acceptance decision, and we've now uploaded the camera ready version. We thank all the reviewers and the action editor for all the invaluable feedback.

---

### Decision · Action_Editors · 2023-04-14

**Recommendation:** Accept as is

**Comment:**

This paper studies personalized treatment of patients. The problem is formulated as a latent bandit in the fixed-confidence pure exploration setting. Each patient type is represented by a latent state. To solve the problem, the state needs to be identified. The authors prove a lower bound on the stopping time of any algorithm for solving the problem. Two algorithms are proposed. One of them is analyzed and an upper bound on its stopping time matches the lower bound. Both algorithms are evaluated empirically.

I like this paper and the reviewers generally support it as well. The leaning reject decision points out that the analyses are on the simpler side. I agree. On the other hand, this is the first paper on latent bandits in pure exploration. The paper is well written and the experiments are also well done. I also wanted to bring

* [Thompson Sampling with a Mixture Prior](https://proceedings.mlr.press/v151/hong22b.html)

to the attention of the authors. This paper generalizes Hong et al. (2020) to imperfect models. It is essentially Thompson sampling with a mixture prior.

**Audience:**

This is a nice paper that extends Hong et al. (2020) to fixed-confidence pure exploration. It will appeal to the bandit community.

**Claims And Evidence:**

Yes. A lower bound and an upper bound on the stopping time of the algorithm. Experimental results.